# Human β-Defensins in Diagnosis of Head and Neck Cancers

**DOI:** 10.3390/cells12060830

**Published:** 2023-03-07

**Authors:** Jenna Kompuinen, Mutlu Keskin, Dogukan Yilmaz, Mervi Gürsoy, Ulvi Kahraman Gürsoy

**Affiliations:** 1Department of Periodontology, Institute of Dentistry, University of Turku, 20520 Turku, Finland; 2Oral and Dental Health Department, Altınbaş University, İstanbul 34147, Turkey; 3Department of Periodontology, Faculty of Dentistry, Sakarya University, Sakarya 54050, Turkey; 4Welfare Division, Oral Health Care, 20101 Turku, Finland

**Keywords:** antimicrobial peptides, human beta-defensins, cancer, diagnosis, immune response

## Abstract

Head and neck cancers are malignant growths with high death rates, which makes the early diagnosis of the affected patients of utmost importance. Over 90% of oral cavity cancers come from squamous cells, and the tongue, oral cavity, and salivary glands are the most common locations for oral squamous cell carcinoma lesions. Human β-defensins (hBDs), which are mainly produced by epithelial cells, are cationic peptides with a wide antimicrobial spectrum. In addition to their role in antimicrobial defense, these peptides also take part in the regulation of the immune response. Recent studies produced evidence that these small antimicrobial peptides are related to the gene and protein expression profiles of tumors. While the suppression of hBDs is a common finding in head and neck cancer studies, opposite findings were also presented. In the present narrative review, the aim will be to discuss the changes in the hBD expression profile during the onset and progression of head and neck cancers. The final aim will be to discuss the use of hBDs as diagnostic markers of head and neck cancers.

## 1. Introduction

Head and neck cancer encompasses a plethora of malignant growths that can manifest in diverse regions of the head and neck. These areas include the lips, oral cavity, different parts of the pharynx, paranasal sinuses, nasal cavity, larynx, and salivary glands [1,2]. Head and neck squamous cell carcinoma is a specific subtype of cancer that arises from the mucosal epithelial cells of the oral cavity, pharynx, and larynx. It represents a significant proportion, around 90%, of all cases of head and neck cancer [3,4]. Possible locations for head and neck squamous cell carcinoma are the tongue, oral cavity, and salivary glands, but the tongue is the most common oral squamous cell carcinoma site [5]. Head and neck squamous cell carcinoma develops through certain phases, including epithelial cell hyperplasia, dysplasia, carcinoma in situ, and eventually develops invasive carcinoma [6]. Oral premalignant lesions, which may lead to oral cancer, are important to monitor in order to detect cancer at an early stage. Premalignant lesions include leukoplakia, erythroplakia, and erythroleukoplakia. 

Nasopharyngeal carcinoma is rare, and its incidence has declined worldwide over the past decades [7]. It arises from the nasopharynx epithelium. The World Health Organization (WHO) categorizes nasopharyngeal carcinoma into three pathological subtypes depending on keratinization and tumor differentiation. Nasopharyngeal carcinoma is associated with poor prognosis. Its symptoms are, for example, nose hemorrhage and headache [8].

The remaining 10% of head and neck cancers includes sarcomas, salivary gland tumors, lymphoma, and melanoma [9]. Only 5% of tumors in head and neck region are salivary gland tumors [10]. Most of them are benign neoplasms, such as pleomorphic adenoma. Salivary gland malignancies are a heterogenous group of tumors consisting of over 20 different histological types [11]. The parotid, submandibular, and sublingual salivary glands are the major paired salivary glands. Indeed, there is also a high number of minor salivary glands in the oral cavity. Mucoepidermoid carcinomas are the most common salivary gland malignancy and 69% of them are diagnosed in the parotid gland [12]. Thus, the parotid is the most common site for salivary gland tumors, followed by the submandibular gland, with the rarest among the major salivary glands being the sublingual gland. However, overall risk of malignancy increases when the tumor is located in small salivary glands, i.e., in the sublingual gland [13]. The site of salivary gland tumor is relevant to prognosis [14]. In particular, those cancers which easily transmit metastases have poor prognosis compared to those without metastases. Risk factors for salivary gland tumors are nutrition, irradiation, and long-standing histologically benign tumor that occurs at youth [10].

According to the 2022 report of the American Cancer Society, the estimated number of new oral cavity and pharynx cancer cases in the USA alone is 1,918,030, which may lead to the death of 609,360 patients [15]. In continuation, being diagnosed with oral cavity and pharynx cancer and at risk of death is twice as high for males as it is for females. On a worldwide scale, GLOBOCAN 2020 estimates that the number of new cases of tumors arising in the lip, oral cavity (377,713), larynx (184,615), nasopharynx (133,354), hypopharynx (84,254), and salivary glands (53,583) will amount to around 830,000, accounting for 4.4% of all new global cancer cases. Additionally, head and neck squamous cell carcinoma, with its diverse biology and genomic heterogeneity, is also responsible for an estimated 420,000 deaths globally, representing 3.4% of all cancer-related deaths [16].

The most common treatments for head and neck cancer are surgery, radiotherapy, and chemotherapy. Usually, the best treatment is surgery in combination with radiotherapy or chemotherapy, if needed, but in nasopharyngeal carcinoma, radiotherapy and chemotherapy is preferred rather than surgery [8]. Treatments can cause a number of side effects to patients and the quality of life can be reduced. Therefore, new methods of cancer treatment need to be studied. For example, herbal drugs [17] and sonodynamic therapy [18] could be future methods for head and neck treatments when they have been investigated more. Herbal drugs can inhibit the progression of disease and can be a safe treatment with less side effects [17]. Sonodynamic therapy using low-intensity-ultrasound with a sonosensitizer, in theory, seems to be effective for cancer [18]. However, more studies are needed about treatment methods.

The aim of this present narrative review is to first give a brief overview of head and neck cancers. Afterwards, human beta-defensins will be described in general. Finally, current evidence on the possible use of human beta-defensins as biomarkers of head and neck cancers will be presented. The pathogenesis, diagnosis, or the treatment of head and neck cancers are not in the scope of this narrative review and readers who are willing to obtain more information on these topics can reap the benefits of well-written reviews [6,19].

## 2. Risk Factors of Head and Neck Cancers

The pathogenesis of head and neck cancer is not yet fully understood. However, it is believed to be a complex disorder that results from a combination of genetic factors, lifestyle choices, and environmental exposures [20,21]. There is a well-established correlation between tobacco and cigarette consumption with head and neck squamous cell carcinoma. The use of other combustible and smokeless tobacco products also heightens the risk of head and neck squamous cell carcinoma, specifically for oral cavity cancers. The combination of smoking and alcohol consumption has a particularly strong effect on the development of head and neck squamous cell carcinoma, with the population attributable risk being 72% [22,23]. In addition to traditional risk factors such as smoking and alcohol consumption, over the last two to three decades, the connection between certain viruses such as human papillomavirus (HPV) and Epstein Barr Virus (EBV) and the development of certain types of head and neck squamous cell carcinoma have also been identified [24].

## 3. Diagnosis of Head and Neck Cancers

Head and neck cancer is a complex and heterogeneous disease that poses a significant diagnostic challenge [25,26]. The diagnostic workup for head and neck cancer typically begins with a comprehensive clinical examination, which is complemented by various biochemical investigations, invasive biopsy, and radiologic imaging [1,27]. Clinical decision making in head and neck oncology is based on various parameters, including the patient’s medical history, performance and nutritional status, comorbidities, physical symptoms, and signs, as well as the results of laboratory tests, radiological imaging, and pathological and molecular investigations [28]. Conventional pathological diagnosis, which uses incisional biopsy of the primary tumor or fine-needle aspiration cytology as sample material, is still the most reliable qualitative diagnosis [29]. There are currently immunohistochemical markers that are routinely performed to confirm a diagnosis, such as anti-Epithelial Membrane Antigen (EMA), p63, p40, or cytokeratins used in squamous cell carcinoma [30]. Human saliva could be an important factor in the diagnosis of head and neck cancer, especially when detecting cancer early. When taking a saliva sample, the patient does not suffer any pain, which is positive compared to invasive sampling. Biomarkers from saliva could also help determine prognosis for the disease [9].

Radiologic imaging plays an important role in the diagnosis and staging of head and neck cancer. Ultrasound is the preferred method for initial evaluation of enlarged cervical lymph nodes, as it allows for assessment of the size, location, composition, and relationship to the great vessels [29,31]. Computed tomography (CT) is commonly used in oncology to assess the primary site of head and neck cancer, nodal disease, and stage of the cancer. It can provide high-resolution multiplanar images for anatomical demonstration and treatment planning [32,33]. Magnetic resonance imaging (MRI) is considered to be the best imaging technique for evaluating certain areas of the head and neck, specifically the upper portion of the neck, such as the nasopharynx, sinuses, oral cavity, and oropharynx. The 18F-fluorodeoxyglucose PET (FDG-PET) is a widely used diagnostic tool in the management of head and neck squamous cell carcinoma, as it has been shown to have a high sensitivity and negative predictive value for identifying small lymph nodes in the neck, making it a valuable tool for staging and treatment evaluation. FDG-PET-CT is currently recommended as part of the staging process for head and neck squamous cell carcinoma if there is evidence of cancer spread beyond the primary site, especially in advanced stages [34,35,36].

### Diagnostic Biomarkers of Head and Neck Cancers

Early diagnosis is crucial for the treatment of head and neck squamous cell cancers, as treatment is most effective when the tumor size at the primary site is lowest and there is the least lymphatic and hematogenous spread. Biomarkers, defined as biological molecules found in blood, body fluids, or tissues, can serve as signs of normal or abnormal processes or conditions and are a potential solution for early diagnosis [37]. Emerging studies on large head and neck squamous cell carcinoma patient cohorts have been carried out to find predictive biomarkers that could help clinicians make accurate early diagnoses, predict clinical outcomes, and provide a reference for individualized immunotherapy for head and neck squamous cell carcinoma patients [38]. However, although various biomarkers have been suggested for head and neck squamous cell carcinoma, few of them have been validated for use in clinical practice [39]. Various surface markers were defined and clinically validated as indicators for progression or treatment response in oral cancers [40].

Some markers have emerged as being fundamental in the diagnosis of tumors, as they have prognostic and therapeutic implications. Several well-known prognostic factors can be easily assessed by immunohistochemistry (IHC), including the presence of mutations of the TP53 tumor-suppressor gene and the cell proliferation marker Ki-67 [29]. Other markers such as cortactin, NANOG, and SOX2 protein expression are frequent in squamous cell carcinoma. Among them, NANOG was proposed as an independent predictor of better clinical outcome in head and neck squamous cell carcinoma. Moreover, according to the same study, the combined expression of NANOG and SOX2 increased the prognostic significance [41]. PD-L1 expression has been studied as a potential predictive biomarker for the response to treatment with immunotherapy in head and neck cancer. Similar to other types of cancer, PD-L1 is a protein that is expressed on the surface of some head and neck squamous cell carcinoma cells, and its expression can regulate the immune system’s response to the cancer cells by binding to the PD-1 receptor on T-cells. Human epidermal growth factor receptor 2 (HER2) is a protein that is expressed on the surface of some cells, including cancer cells, and it plays a role in the growth and survival of cancer cells. HER2 overexpression, which is defined as an abnormal increase in the number of HER2 receptors on the surface of cancer cells, has been observed in a subset of head and neck squamous cell carcinomas, as well as in other types of cancer, such as breast cancer. HER2 overexpression can be used as a diagnostic biomarker for head and neck squamous cell carcinoma, as well as a prognostic marker. Studies have shown that HER2 overexpression is associated with more aggressive tumor behavior and a poorer prognosis in head and neck squamous cell carcinoma patients [42]. HPV tumor status is also a prognostic factor for head and neck cancer; patients with HPV-positive squamous cell carcinoma have a better overall prognosis compared to patients with HPV-negative squamous cell carcinoma. Prognosis for head and neck cancer depends on anatomic site, stage, and HPV tumor status. Lip and oral cavity cancers have the highest incidence and mortality worldwide [43,44]. There are other potential biomarkers that have been studied in head and neck cancer, including EGFR, KRAS, PIK3CA, and p16INK4a protein [45,46,47,48,49]. However, their clinical utility is still under investigation and more research is needed to fully understand their role in the diagnosis and management of head and neck cancer.

## 4. Immune Response Regulation in Head and Neck Cancers

The human immune system plays a crucial role in the initiation and progression of cancer, particularly in head and neck squamous cell carcinoma [50]. The immune system, through surveillance and elimination, is responsible for recognizing self versus non-self and protecting the body from diseases of exogenous and endogenous origins [50]. The presence, polarizations, and activities of immune cells, primarily dendritic cells, T-lymphocytes, B-cells and plasma cells, some natural killer cells (NK), macrophages, and eosinophils impact the onset and progression of head and neck squamous cell carcinoma. However, head and neck squamous cell carcinoma is a highly immunosuppressive malignancy due to multiple mechanisms including induction of immune tolerance, local immune evasion, and disruption of T-cell signaling. Additionally, head and neck squamous cell carcinoma has the ability to evade the immune surveillance system through various mechanisms. These mechanisms include the modulation of inflammatory cytokines, the suppression of cytotoxic CD8 lymphocytes, the downregulation of antigen-processing machinery, the generation of specific inhibitory lymphocytes, and the expression of immune checkpoint ligands and/or their receptors [51,52]. These cellular mechanisms, together with the upregulation of inhibitory checkpoint receptors that can inhibit normal T-cell activation inside the tumor, allow the tumor to resist against cytotoxic T-cells and continue to grow [53].

### 4.1. Oral Cavity and Human β-Defensins

The oral cavity is a unique part of the human body where continuous communication with the external environment is observed. It hosts niches for both commensal and pathogenic bacteria [54]. The first source of immunity in the oral cavity is provided by the epithelial cells of the oral mucosa. It has been known that the epithelium structure of the oral cavity does not simply function as a passive barrier between intra- and extraoral environments [55]. For instance, oral epithelial tissues synthesize a chemical barrier in the form of antimicrobial peptides that are effective against oral pathogens in a multifunctional manner [56]. Human antimicrobial peptides have positive charges and possess dynamic structural properties due to the variable biochemistry of amino acid residues [57]. This structural variability confers their function and their role in immune defense [58]. Human defensins are among the major antimicrobial peptides which are well-defined in the oral cavity. The common structure of these small, cationic peptides consists of a β- sheet structure and three disulphide bonds [59]. Human oral defensins can be classified into two groups: α- and β-defensins. The distinctions in the connecting patterns of three disulfide bonds and the spacing of cysteine determines the type of defensin [60].

Human β-defensins (hBDs) are expressed in various epithelial tissues in the human body, including the oronasal cavity, gingiva, dental pulp, tongue, salivary glands, and mucosa [61,62]. After being expressed, these antimicrobial peptides can be detected in gingival tissues, saliva, gingival crevicular fluid, and nasal secretions [63,64]. As well as epithelium, it has been also demonstrated that various cell types including monocytes, macrophages, monocyte-derived dendritic cells, odontoblasts, and keratinocytes can express hBDs [65,66]. Genomic studies have identified 28 different hBDs in the human body; however, only 4 (hBD1–4) have been detected in gingival epithelium [67,68]. In gingival epithelium, hBDs are localized and expressed in stratified epithelium, whereas no expression was detected in junctional epithelium [63]. hBDs present various modes of expression, which occurs either constantly, or after stimulation with bacterial lipopolysaccharides, inflammatory mediators, or neoplastic lesions [69]. For instance, the expression of hBD-1 is constitutive in epithelial tissues, while hBD-2 and hBD-3 are inductively expressed by the aforementioned stimulants [70]. Studies revealed high interindividual variation in the expression of both genes and proteins for hBDs in oral tissues and fluids [60,63,71].

hBDs are multifunctional peptides that function in a coordinated manner [72]. The first defined function of these peptides was their antimicrobial properties. Antimicrobial activities of hBDs show broad-spectrum antibacterial, antifungal, and antiviral activities through depolarizing and permeabilizing microbial cell membranes due to their cationic charges [73]. The antimicrobial activities of hBDs show great variation in the oral cavity compared with other parts of human body [59]. hBD-3 has the most potent activity and the highest positive charge (+11) among gingival hBDs [74]. It is effective against both Gram-positive and Gram-negative bacterial species, as well as Candida albicans, while hBD-1 is weakly against Gram-negative bacteria [75,76]. Local salt concentration is the major determinant of the antimicrobial activities of these peptides. Decreased antimicrobial activity of hBDs was observed in saliva compared to gingival tissues; however, it is unclear if it is related to the salivary salt concentrations, as salivary salt concentrations are generally low [77]. Bacteria and their metabolites can induce expression of hBDs via either toll-like receptors or independent signaling pathways [78]. However, some periodontopathogens including Treponema species are resistant to hBD-1, -2, and -3 and enzymatic degradation of hBDs by proteases was also demonstrated [79,80]. Besides their well-known antimicrobial activity, hBDs play essential roles between innate and adaptive immunity to establish homeostasis in the human body. For instance, they engage the CCR6 receptor on selected immune cells, such as monocytes, macrophages, and mast cells, evoking a chemotactic response [81]. hBD 1–3 can also function directly as chemokines for dendritic cells in combination with recruiting T-cells [82]. This function of hBDs may be a link between innate and adaptive immune activation. In addition, cell maturation and the antigen presentation activity of dendritic cells are stimulated by hBD-1 and -2 [83]. hBDs not only function as antimicrobial and immune regulators, but they may also contribute to the wound healing of the periodontium. It has been demonstrated that both hBD-2 and -3 increase the keratinocyte migration and proliferation by the induction of STAT proteins and epidermal growth factor, respectively, which can be important in the re-epithelization phase of the repair [84,85].

The levels of hBDs in oral fluids and tissues are modulated by oral infectious diseases (periodontal diseases, caries, pulpal infections) or systemic diseases, or conditions such as diabetes mellitus (DM) and pregnancy, and oral cancers [60,63,86,87,88]. In the literature, no consensus exists regarding the relation between hBDs and the extent of periodontal inflammation. Protein and mRNA levels of hBDs in oral biological fluids and tissues have been previously found to be elevated, steady, or suppressed in participants with gingivitis or periodontitis [60,69,71]. The dysbiotic and inflammatory nature of the disease, increased host- and bacteria-derived enzymes, genetic polymorphisms, and disrupted epithelial structure of the periodontium due to disease may lead to the controversial findings of the literature [70]. According to the results from our group, overexpression of hBD-2 and hBD-3 in gingival tissues of Type 2 DM patients and altered salivary levels of hBDs in Type 1 DM can partly explain why diabetic patients are more prone to periodontal diseases [63,89,90]. In human dental pulp tissues, it was demonstrated that not only oral keratinocytes at the epithelial surface but also odontoblasts express hBDs [91]. The gene expression levels of hBD-1 and -2 were increased in inflamed pulps while no change was detected in hBD-3 [87]. Jurczak et al. (2015) indicated a significant relationship between early childhood caries and salivary hBD-2 levels, and they recommended this antimicrobial peptide as a potential disease progression biomarker [92]. Finally, age may act as a confounding factor in the relation between oral hBD levels and the extent of periodontal destruction [93].

### 4.2. Oral hBDs in Head and Neck Cancers

The idea that hBDs can regulate tumor growth and microenvironment by their multifunctional properties is more than 2 decades old [94]. Yet, the gene and protein expression profiles of hBDs are dependent on cancer type and its anatomical location [88]. Today it is still unclear if hBDs act as tumor suppressors or promoters; hBDs can manipulate the tumor microenvironment and support tumor growth, but also can exhibit direct cytotoxic activity toward cancer cells or by activating antitumor immunity [72,88]. Indeed, the question whether hBDs act as tumor-suppressor genes or proto-oncogenes has not been answered [88]. hBDs can modify tumor cells’ capacity and may favor their proliferation, migration, and invasion to adjacent tissues.

Overexpression of hBD-2 in esophageal cancer is able to promote cell growth of KYSE-150 cell lines through the NF-κB pathway [95]. hBD-3 enhances cancer metastasis, and this effect can be blocked by inhibiting EGFR or neutralizing TLR4 in SCC-25 cells [96]. These findings suggest that hBD-3 may play a role in the progression of OSCC. hBD-3 also can protect cisplatin-mediated apoptosis of SCC of the head and neck cells through the PI3/Akt pathway, indicating a role of hBD-3 in promoting cancer survival cell [97]. It is important to establish a tumor-associated microenvironment for the growth and progression of the tumors. The multifunctional characteristics of hBDs directly or indirectly may help to establish this environment. For instance, hBDs can promote angiogenesis and atypical activation of angiogenesis is a significant sign of cancer [98]. On the other hand, overexpression of hBD-3 recruits monocytes from the peripheral blood and regulates the infiltration of tumor-associated macrophages which favors a tumor-associated environment [99]. Figure 1 illustrates the impact of hBDs on tumor cells and environment.

Until now, various studies evaluated the clinical changes in the hBD levels in head and neck cancer patients [94,95,96,100,101,102,103,104,105,106,107,108,109,110,111]. Their findings are summarized in Table 1.

According to the available evidence, there is no generalized expression pattern of hBDs in patients with oral cancers. Han et al. (2014), Wang et al. (2014), Pantelis et al. (2009), Kesting et al. (2012), and Wenghoefer et al. (2008) showed that hBD-1 expression is reduced in OSCC tissue samples [100,101,103,105,106]. Han et al. (2014) also studied precancerous lesions and demonstrated that hBD-1 expression decreases from oral precancerous lesions to OSCC [100]. hBD-1 promotes cancer cell apoptosis, and it also suppresses tumor migration and invasion of OSCC. It is demonstrated that hBD-1 expression in OSCC tissue is higher in patients without lymph node metastasis, and it is associated with better prognosis of patients. Thus, hBD-1 expression is excellent predictor of cancer-specific survival of OSCC patients, and it is associated with better prognosis [100].

hBD-2 expression is usually limited to the superficial layers of healthy oral mucosa. Like hBD-1 expression, hBD-2 expression is also reduced in OSCC, in salivary gland tumors, and during the malignant transformation of tonsillar carcinoma [94,101,103,107]. Indeed, there is a positive correlation between hBD-2 levels and tumor differentiation in OSCC, and hBD-2 is associated with lymph node metastasis [101]. In contrast to the findings, which demonstrated decreased hBD-2 levels in tumors, there are also studies that demonstrated a higher number of hBD-2 positive cells and hBD-2 RNA expressions in tissues with squamous cell carcinoma, in comparison to healthy tissues [95,108,111].

Contrary to the hBD-1 and hBD-2 protein expression profiles, hBD-3 levels are generally found to be high in most of the studies investigating OSCC tissues [96,104,106,109]. There are contradictory findings as well, which indicates decreased hBD-3 levels in OSCC tissue samples, in comparison to healthy oral mucosa [101]. Of course, the localization of the healthy tissue collection site may have an importance, as the control samples of the Wang et al. (2008) study were collected from impacted third molar surgical extractions [101]. Finally, translocation of hBD-3 in OSCC tissue samples may explain the differences in results, as elevated hBD-3 was previously localized to the cytoplasm of malignant epithelium cells and in inflammatory cells [96].

### 4.3. Human β-Defensins in Diagnosis of Head and Neck Cancers

Detecting tumor growth at its early phases is the primary goal of cancer diagnosis. An ideal biomarker must objectively indicate the normal biological process, the pathogenic process, or the response to therapeutic intervention [112]. Bearing this in mind, the significant shifts in hBD levels in head and neck cancers may allow them to be considered as biomarker candidates. Yet, the available evidence is highly limited with oral squamous cell carcinoma and the protein expression profiles of hBDs differ from each other. A common finding is the decreased hBD 1–2 levels in cancer tissues in comparison to non-tumorigenic tissues. While hBD levels are prone to proteolytic activity, decrease in hBD-1 and hBD-2 in oral squamous cell carcinoma cannot be solely related to post-transcriptional degradation, as hBD-3 levels tend to increase simultaneously. Indeed, elevated hBD-3 levels in oral squamous cell carcinoma may indicate the function of this peptide in cell proliferation. Moreover, EGFR, which induces the expression of hBD-3, is overexpressed in many cancer types, including OSCC [96]. Finally, the available evidence is only limited to tissue biopsies; therefore, the majority of the studies used immunohistochemistry as the method of detection. Considering that biopsy collection may have its own risks, studies on salivary (or, less preferably, serum) levels of hBDs are necessary to suggest the possible use of hBDs as biomarkers of head and neck cancers.

## 5. Conclusions and Future Perspectives

The present review made an effort to collect the evidence on the use of hBDs as diagnostic biomarkers of head and neck cancers. According to the current literature, hBD-1 and hBD-2 are downregulated, while hBD-3 is increased in head and neck cancers; however, this information is limited to oral squamous cell carcinoma. Indeed, the expression and regulation profiles of hBDs in cancers are cancer-specific and they interact differently with receptors on cells, which can be explained by the promiscuous nature of these small peptides [88,113]. Therefore, creating general principles from the detailed facts as to the fate of the β-defensins in cancer must be avoided. Considering the high death rates, there is a significant need of new diagnostic biomarkers for head and neck cancers which allow for early diagnosis and treatment. Defining the expression profiles, cellular receptors, and the biological roles of these small antimicrobial peptides in head and neck cancers may allow researchers to propose the potential use of hBDs as diagnostic, prognostic, or therapeutic agents.

## Figures and Tables

**Figure 1 cells-12-00830-f001:**
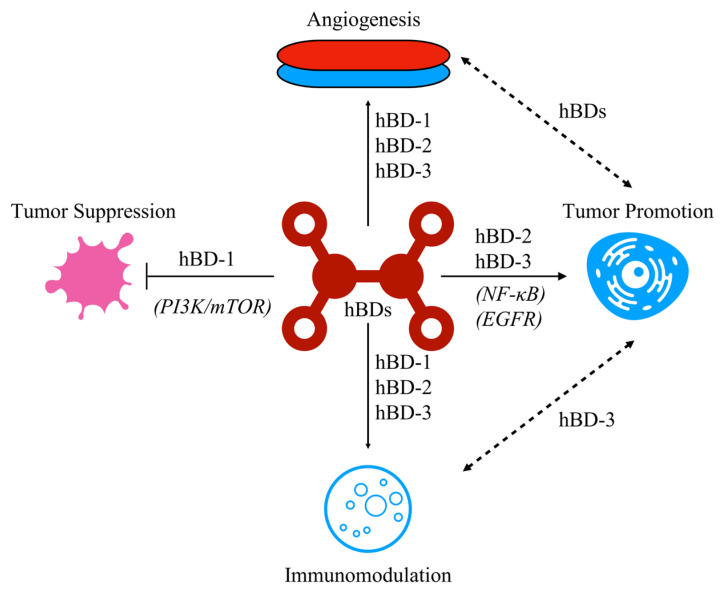
The multifunctional properties of human beta-defensins (hBDs) in regulating tumor growth and microenvironment. The diagram illustrates the conflicting roles of hBDs in cancer progression, and their context-dependent effects. The italicized text indicates possible pathways.

**Table 1 cells-12-00830-t001:** Summary of the clinical studies evaluating the hBD levels in head and neck cancer patients.

Author, Year	Cancer Type	Study Population	Detection Methods	Findings
Han et al., 2014 [100]	OSCC	30 OSCC; 17 oral leukoplakia; 15 controls	IHC	HBD-1 is reduced in OSCC compared to oral leukoplakia and is lower in OSCC with lymph node metastasis
Wang et al., 2014 [101]	OSCC	60 OSCC; 30 oral leukoplakia; 15 controls	IHC	hBD-1, -2, and -3 is reduced in OSCC. hBD-2 and -3 correlated with lymph node metastasis of OSCC
Shuyi et al., 2011 [96]	OSCC	79 OSCC; 79 controls	IHC RT-qPCR	Protein and mRNA expressions of hBD-3 are elevated in OSCC
Kesting et al., 2009 [102]	OSCC	45 OSCC; 16 controls	IHC RT-PCR	hBD-3 mRNA is elevated in OSCC
Wenghoefer et al., 2008 [103]	OSCC	5 OSCC; 5 irritation fibroma; 5 leukoplakia; 5 controls	RT-PCR	Gene expressions of hBD-1 and -2 are reduced and hBD-3 is elevated in OSCC
Hussaini et al., 2006 [104]	OSCC	29 OSCC; 23 controls	IHC	hBD-3 is elevated in stage II and III OSCC compared to stage I and IV
Abiko et al., 1999 [94]	OSCC	4 OSCC	RT-PCR	hBD downregulation in SCC
Pantelis et al., 2009 [105]	Pleomorphic adenoma	5 pleomorphic adenoma; 5 adenoma-adjacent normal tissue; 5 chronic sialadenitis; 5 controls	RT-PCR IHC	hBD-1 expression levels are decreased in pleomorphic adenomas compared to controls
Shi et al., 2014 [95]	Esophageal SCC	58 SCC; 50 Control	IHC	IHC scores of hBD-2 are higher in tumors than in normal tissues
Kesting et al., 2012 [106]	Salivary gland tumor	7 malign salivary gland neoplasm; 10 benign salivary gland neoplasm; 41 controls	qrt-PCR IHC	The expression of hBD-1, -2, and -3 are reduced in salivary gland tumor tissue samples
Pacova and Martinek 2009 [107]	Tonsillar carcinoma	50 nasal polyposis; 11 chronic tonsillitis; 17 tonsillar carcinoma	IHC	hBD-1, -2, and -3 secretions are decreased during malignant transformation
Chong et al., 2006 [108]	Recurrent respiratory papillomatosis	15 RPP; 10 controls	RT-PCR IHC	hBD-1, -2, and -3 mRNAs were higher in RPP compared to healthy oral mucosal tissues
Wenghoefer et al., 2008 [109]	Salivary gland tumors	7 malign salivary gland tumor; 7 benign salivary gland tumor; 7 controls	IHC	hBD-1 migrates into the nucleus of malignant salivary gland tumors
Lee et al., 2022 [110]	Tongue SCC	23 tongue SCC with no regional metastasis; 12 tongue SCC with positive regional metastasis	mRNA sequencing IHC	Reduced expression of defensin related genes are associated with regional metastasis to the neck in early-stage tongue cancer
Hoppe et al., 2016 [111]	Head and neck tumors	15 head and neck tumors; 15 healthy gingiva	IHC RT-PCR	hBD-1, -2, and -3 expression are increased in OSSC.

## Data Availability

Not applicable.

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
