# Peer review of "Human β-Defensins in Diagnosis of Head and Neck Cancers"

_cells, 2023, doi:10.3390/cells12060830_

Round 1

Reviewer 1 Report

Head and neck cancers have a significant impact on human morbidity and mortality. This review by Kompuinen and colleagues deals with a possible correlation between beta-defensin levels and the occurrence of head and neck cancers. Diagnosis of head and neck cancer is not trivial. Therefore, the chosen topic of interest due to its implications, i.e. the possible use of human beta-defensins as biomarkers for early diagnosis of head and neck cancers.

General comments

1.      It is not always clear whether the hBDs  described were observed in the tumor cells or in the tumor surroundings, stroma, or inflammatory cells?

Specific comments

Line 16: please correct “findingswere

Lines 16-18: “In the present narrative review, the aim will be to discuss the changes in hBD expression profile during the onset and prognosis of head and neck cancers.” Not sure that “prognosis” is a fitting term in this context (perhaps outcome?).

Lines 28-29: “it was projected that in 2022 to be a staggering…”.

“to be” doesn’t sound correct, consider replacing or deleting.

Line 69: Consider changing “Cervical masses”.  Perhaps “enlarged cervical lymph nodes”.

Lines 100-102: I find it hard to understand the sentence “Some markers such as cortactin, NANOG and SOX2 protein expression are frequent in squamose cell carcinoma; combined expression of both proteins showed the highest survival rates”.

Line 139: “Tolerance to cytotoxic T-cells” do you mean induction of T cell tolerance?

Line 160: “They can migrate into the secretions”. “Migrate” sound more relevant to an object that has motility, to the ability to govern movement, not for a passive molecule.

Line 177: “….variation in oral cavity…” please add “the” to make “.variation in the oral cavity…”.

Lines 182-183: “Decreased antimicrobial activity of hBDs was observed in saliva compared to gingival tissues due lower electrostatic interactions [63].To my understanding, reference [63] states that lower electrostatic interactions are probably not the cause of the decreased antimicrobial activity of hBDs in saliva as follows: “…antibacterial activities of hBD 1-3 and hCAP18/LL-37 by 20–50% in in vitro conditions (Mineshiba et al., 2003). This in vitro effect is generally explained by the salt concentration of saliva. However, this is probably unlikely because of the low salt concentrations in saliva.” Please elaborate.

Line 201: I suggest to modify “In literature” to “In the literature”.

Lines 222-224: “The common approach is that the hBDs contribute to the cancer development by acting as tumor suppressor genes, exhibiting direct cytotoxic activity toward cancer cells or by activating anti-tumor immunity”. This sentence needs to be clarified. First, do hBDs contribute to cancer development, or do they normally suppress tumor development?

Second, regarding hBDs as tumor suppressor genes, are the hBDs genes mis-regulated/mutated in the cancer cells themselves (indicating that their genes might be tumor suppressors, or protooncogenes), or are they mis-regulated in the tumor environment by neighboring cells? Reference 74 might be also fitting here as a more updated citation.

Line 233: Please check if there is a space missing in “…90].Han…”.

Line 242: remove the “.” From “…mors. and…”

Lines 243-244: “there is correlation between hBD-2 levels and tumor differentiation in OSCC and hBD-2 is associated with lymph node metastasis”. Is this a positive correlation or an inverse one?

Lines 245-246: Please rephrase “that the number of hBD-2 positive cells and hBD-2 RNA expressions are elevated in tissues with squamous cell carcinomato make it more coherent to the reader.

Line 250: pleas add a missing space in “Ofcourse”.

Line 258: “Or” can be deleted from  or the patho-…”.

Lines 260-261: “disappearance of hBDs in head and neck cancers may allow them to be considered as biomarker candidates”. Do you intend disappearance of all hBDs or only hBD-1 and 264 hBD-2?

Author Response

Head and neck cancers have a significant impact on human morbidity and mortality. This review by Kompuinen and colleagues deals with a possible correlation between beta-defensin levels and the occurrence of head and neck cancers. Diagnosis of head and neck cancer is not trivial. Therefore, the chosen topic of interest due to its implications, i.e. the possible use of human beta-defensins as biomarkers for early diagnosis of head and neck cancers.

General comments

  1. It is not always clear whether the hBDs described were observed in the tumor cells or in the tumor surroundings, stroma, or inflammatory cells?

Authors’ response: Literature findings cover both tumor and tumor-surrounding tissues. This is now described in the present manuscript.

Specific comments

Line 16: please correct “findingswere”

Authors’ response: Sentence is corrected (Page 1, line 20)

Lines 16-18: “In the present narrative review, the aim will be to discuss the changes in hBD expression profile during the onset and prognosis of head and neck cancers.” Not sure that “prognosis” is a fitting term in this context (perhaps outcome?).

Authors’ response: Sentence is corrected as “… during the onset and progression of head and neck cancers.” (Page 1 line 21-22)

Lines 28-29: “it was projected that in 2022 to be a staggering…”.

“to be” doesn’t sound correct, consider replacing or deleting.

Authors’ response: Sentence is rewritten (Page 1, lines 59-62)

Line 69: Consider changing “Cervical masses”.  Perhaps “enlarged cervical lymph nodes”.

Authors’ response: Sentence is corrected (Page 2, lines 117-118)

Lines 100-102: I find it hard to understand the sentence “Some markers such as cortactin, NANOG and SOX2 protein expression are frequent in squamose cell carcinoma; combined expression of both proteins showed the highest survival rates”.

Authors’ response: Sentence is rewritten (Page 2 line 150, Page 3 lines 151-154)

Line 139: “Tolerance to cytotoxic T-cells” do you mean induction of T cell tolerance?

Authors’ response: Sentence is rewritten (Page 4 lines 191-194)

Line 160: “They can migrate into the secretions”. “Migrate” sound more relevant to an object that has motility, to the ability to govern movement, not for a passive molecule.

Authors’ response: Sentence is rewritten (Page 5, lines 213-214)

Line 177: “….variation in oral cavity…” please add “the” to make “.variation in the oral cavity…”.

Authors’ response: Corrected (Page 5, line 213-214)

Lines 182-183: “Decreased antimicrobial activity of hBDs was observed in saliva compared to gingival tissues due lower electrostatic interactions [63].” To my understanding, reference [63] states that lower electrostatic interactions are probably not the cause of the decreased antimicrobial activity of hBDs in saliva as follows: “…antibacterial activities of hBD 1-3 and hCAP18/LL-37 by 20–50% in in vitro conditions (Mineshiba et al., 2003). This in vitro effect is generally explained by the salt concentration of saliva. However, this is probably unlikely because of the low salt concentrations in saliva.” Please elaborate.

Authors’ response: Sentence is rewritten (Page 5, lines 235-238)

Line 201: I suggest to modify “In literature” to “In the literature”.

Authors’ response: Corrected (Page 6, line 256)

Lines 222-224: “The common approach is that the hBDs contribute to the cancer development by acting as tumor suppressor genes, exhibiting direct cytotoxic activity toward cancer cells or by activating anti-tumor immunity”. This sentence needs to be clarified. First, do hBDs contribute to cancer development, or do they normally suppress tumor development?

Second, regarding hBDs as tumor suppressor genes, are the hBDs genes mis-regulated/mutated in the cancer cells themselves (indicating that their genes might be tumor suppressors, or protooncogenes), or are they mis-regulated in the tumor environment by neighboring cells? Reference 74 might be also fitting here as a more updated citation.

Authors’ response: Sentence is rephrased (Page 6 lines 276-282)

Line 233: Please check if there is a space missing in “…90].Han…”.

Authors’ response: Corrected

Line 242: remove the “.” From “…mors. and…”

Authors’ response: Corrected (Page 10 line 330)

Lines 243-244: “there is correlation between hBD-2 levels and tumor differentiation in OSCC and hBD-2 is associated with lymph node metastasis”. Is this a positive correlation or an inverse one?

Authors’ response: Corrected (Page 10 line 331)

Lines 245-246: Please rephrase “that the number of hBD-2 positive cells and hBD-2 RNA expressions are elevated in tissues with squamous cell carcinoma” to make it more coherent to the reader.

Authors’ response: Sentence is rewritten (Page 10 lines 332-335)

Line 250: pleas add a missing space in “Ofcourse”.

Authors’ response: Corrected (Page 10 line 339)

Line 258: “Or” can be deleted from  “or the patho-…”.

Authors’ response: Removed

Lines 260-261: “disappearance of hBDs in head and neck cancers may allow them to be considered as biomarker candidates”. Do you intend disappearance of all hBDs or only hBD-1 and 264 hBD-2?

Authors’ response: Conclusion is rewritten (Page 8, line 364-375)

Reviewer 2 Report

Dear Editor, 

Authors provided an interesting title of review article but whole manuscript is poor and it needs to be improved. There are some comments which should be followed before next steps; -It would be better to explain more about the aim, method, achievements and so on, abstract is so short and needs to be improved

- Keywords are not suitable and should be revised and add more keywords

- Introduction and other sections are so poor about head and neck cancer and I suggest you the below references: -Hajmohammadi E, Ghahremanie S, Alam M, Abbasi K, Mohamadian F, Khayatan D, Rahbar M. Biomarkers and common oral cancers: Clinical trial studies. JBUON. 2021;26(6):2227-37. -https://pubmed.ncbi.nlm.nih.gov/34486685/ (https://www.europeanreview.org/article/26522) -https://pubmed.ncbi.nlm.nih.gov/32135187/ (https://doi.org/10.1016/j.lfs.2020.117483) -https://www.sciencedirect.com/science/article/abs/pii/S0014299920307494 (https://doi.org/10.1016/j.ejphar.2020.173657)

-Introduction should be redesigned and the current form is not acceptable. 

-Diagnostic Biomarkers of Head and Neck Cancers section should be more clarify

-Human β-Defensins in Diagnosis of Head and Neck Cancers section is so poor and needs to be more explained.

-it would be better to use figures in your manuscript for better understanding. 

-Conclusion is not useful, authors should rewrite it.

Best, 

Author Response

Authors provided an interesting title of review article but whole manuscript is poor and it needs to be improved. There are some comments which should be followed before next steps; -It would be better to explain more about the aim, method, achievements and so on, abstract is so short and needs to be improved

- Keywords are not suitable and should be revised and add more keywords

Authors’ response: Corrected

- Introduction and other sections are so poor about head and neck cancer and I suggest you the below references: -Hajmohammadi E, Ghahremanie S, Alam M, Abbasi K, Mohamadian F, Khayatan D, Rahbar M. Biomarkers and common oral cancers: Clinical trial studies. JBUON. 2021;26(6):2227-37. -https://pubmed.ncbi.nlm.nih.gov/34486685/ (https://www.europeanreview.org/article/26522) -https://pubmed.ncbi.nlm.nih.gov/32135187/ (https://doi.org/10.1016/j.lfs.2020.117483) -https://www.sciencedirect.com/science/article/abs/pii/S0014299920307494 (https://doi.org/10.1016/j.ejphar.2020.173657)

Authors’ response: Introduction is significantly expanded and given four citations that belong to the Hajmohammadi group are implemented.

As the present manuscript is an invited review, which was planned to present only the available evidence on the use hBDs in head and neck cancer diagnosis, the pathogenesis, current diagnosis, or the treatment of head and neck cancers are not in the scope of the present review. These sections are not widely described in the current manuscript and the scope is clarified.

-Introduction should be redesigned and the current form is not acceptable.

Authors’ response: Introduction is rewritten.

- Diagnostic Biomarkers of Head and Neck Cancers section should be more clarify

Authors’ response: Corrected

-Human β-Defensins in Diagnosis of Head and Neck Cancers section is so poor and needs to be more explained.

Authors’ response: The clinical evidence is limited. Therefore only available evidence is presented and discussed.

-it would be better to use figures in your manuscript for better understanding.

Authors’ response: A figure is implemented

-Conclusion is not useful, authors should rewrite it.

Authors’ response: Corrected

Round 2

Reviewer 2 Report

Dear Editor,

The revised manuscript is acceptable for publication.

Best,

Author Response

Thank you.